# Djulis (*Chenopodium formosanum*) and Its Bioactive Compounds Protect Human Lung Epithelial A549 Cells from Oxidative Injury Induced by Particulate Matter via Nrf2 Signaling Pathway

**DOI:** 10.3390/molecules27010253

**Published:** 2021-12-31

**Authors:** Chin-Chen Chu, Shih-Ying Chen, Charng-Cherng Chyau, Shu-Chen Wang, Heuy-Ling Chu, Pin-Der Duh

**Affiliations:** 1Department of Anesthesiology, Chi-Mei Medical Center, Tainan 710402, Taiwan; chinchen.chu@gmail.com; 2Department of Health and Nutrition, Chia Nan University of Pharmacy and Science, Tainan 71710, Taiwan; shihying@mail.cnu.edu.tw; 3Research Institute of Biotechnology, Hungkuang University, 34 Chung-Chie Road, Shalu County, Taichung 43302, Taiwan; ccchyau@hk.edu.tw; 4Department of Food Science and Technology, Chia Nan University of Pharmacy and Science, 60 Erh-Jen Road, Section 1, Pao-An, Jen-Te District, Tainan 71710, Taiwan; shujen@mail.cnu.edu.tw (S.-C.W.); chuheuy@mail.cnu.edu.tw (H.-L.C.)

**Keywords:** A549 cells, bioactive compounds, djulis (*Chenopodium formosanum*), oxidative injury, particulate matter (PM), signaling pathway

## Abstract

The protective effects of water extracts of djulis (*Chenopodium formosanum*) (WECF) and their bioactive compounds on particulate matter (PM)-induced oxidative injury in A549 cells via the nuclear factor-erythroid 2-related factor 2 (Nrf2) signaling were investigated. WECF at 50–300 µg/mL protected A549 cells from PM-induced cytotoxicity. The cytoprotection of WECF was associated with decreases in reactive oxygen species (ROS) generation, thiobarbituric acid reactive substances (TBARS) formation, and increases in superoxide dismutase (SOD) activity and glutathione (GSH) contents. WECF increased Nrf2 and heme oxygenase-1 (HO-1) expression in A549 cells exposed to PM. SP600125 (a JNK inhibitor) and U0126 (an ERK inhibitor) attenuated the WECF-induced Nrf2 and HO-1 expression. According to the HPLC-MS/MS analysis, rutin (2219.7 µg/g) and quercetin derivatives (2648.2 µg/g) were the most abundant bioactive compounds present in WECF. Rutin and quercetin ameliorated PM-induced oxidative stress in the cells. Collectively, the bioactive compounds present in WECF can protect A549 cells from PM-induced oxidative injury by upregulating Nrf2 and HO-1 via activation of the ERK and JUN signaling pathways.

## 1. Introduction

Air pollution is the largest single environmental risk for human health. The World Health Organization (WHO) reported that an estimated seven million people worldwide are killed by air pollution every year. Moreover, nine out of ten people breathe air that contains high levels of pollutants that exceed the WHO guideline limits [1]. Airborne particulate matter (PM) varies greatly in terms of its physical and chemical composition, source, space, and particle size [2]. PM10 (<10 µm) and PM 2.5 (<2.5 µm) particles are currently the main concern because their molecular size is small enough to penetrate deep into the lungs, potentially causing serious health risks [3]. The main sources of PM10 and PM2.5 substances in the air are road traffic emissions, biomass burning-related sources, uncovered soil and mining operations, industrial waste, and so on [4]. The typical chemical constituents of PM include sulfates, nitrates, other inorganic ions such as ions of sodium, potassium, calcium, and chloride, organic and elemental carbon, metals, metalloids, and polycyclic aromatic hydrocarbons (PAH), as well as allergens and microbial compounds [5]. Many compounds of PM attached to black carbon are currently considered to be responsible for adverse health effects [6].

Numerous studies have shown that exposure to particulate matter less than 2.5 µm in diameter is associated closely with adverse health effects, such as cardiovascular disease, atherosclerosis, the induction of diabetes mellitus, adverse birth outcomes [6], the incidence and exacerbation of asthma, chronic obstructive pulmonary disease [7], endothelial dysfunction, the induction of inflammation, and a decline in lung function [6]. Although the mechanisms of action of the negative effects of PM on human health have not been fully elucidated, oxidative stress may be one of the important mechanisms playing a major role in causing toxicity in living cells and tissues [8]. The oxidative stress due to particulate air pollution exposure induces a series of reactions to biomacromolecules, such as lipids, proteins, and deoxyribonucleic acid (DNA), affecting their structure and biofunction and thereby leading to the impairment of target cells and tissues [6]. The oxidative stress induced by PM may arise from: (1) altered mitochondrial function or suppressed nicotinamide adenine dinucleotide phosphate (NADPH) oxidase; (2) the direct generation of reactive oxygen species (ROS) from the surfaces of soluble compounds present in PM; and (3) the activation of inflammatory cells capable of generating ROS and reactive nitrogen species (RNS), as well as oxidative damage [9]. In addition, oxidative damage by PM exposure may result from an imbalance of antioxidant defense and pro-oxidant processes, which causes increased exposure to oxidants or the presence of impaired antioxidant defenses [9]. Therefore, one of the important strategies for preventing or reducing oxidative stress is to activate the antioxidant defense system and to prevent cells/tissues from generating ROS and RNS.

Djulis (*Chenopodium formosanum*) is an important traditional crop and is one of the ingredients of the local wine brewed by aboriginal people in Taiwan. Djulis, which contains green or purple stripes, is also known as rainbow rice. Recently, djulis has been considered as a potential crop due to its high nutritional value. Therefore, djulis has received much attention in the use of functional foods. Recently, the biological effects of djulis have been investigated. For example, our previous studies showed that djulis and its bioactive compounds offer hepatoprotection [10], the inhibition of hyperglycemia and hyperlipidemia [11], and antiadipogenic [12], antihypertension [13], and anticancer activity [14]. Furthermore, djulis has antioxidant and antidiabetic effects and protects the skin against UV-induced damage [15,16]. In addition, phytochemicals are plant-derived small molecules that possess multifunctional effects. It is worth mentioning that djulis is rich in phytochemicals, such as rutin, quercetin, kaempferol, and betanin, and significantly inhibits oxidative stress [14]. The results from Romieu’s report showed that air pollution exposure results in increased oxidative stress [9]. Moreover, evidence from previous studies has contributed to the hypothesis that PM exposure could induce considerable oxidative stress and systemic inflammation [6]. On the other hand, higher flavonoid intake could reduce oxidative stress and attenuate the adverse health effects induced by PM exposure [17]. That is to say, the bioactive compounds present in natural products and plants may display a significant inhibitory effect against PM-induced oxidative damage in cells. Given that djulis has shown multifunctional potentials, as mentioned above, it is possible that the oxidative stress induced by PM may be significantly inhibited by djulis and its bioactive compounds. However, no results have been reported so far on the effectiveness of djulis in regulating PM-induced oxidative damage. Therefore, the aim of this study is to explore the effects of djulis and its bioactive compounds on regulating PM-induced oxidative stress damage and to elucidate the antioxidant pathways involved.

## 2. Results

### 2.1. Effect of WECF on PM-Induced Damage in A549 Cells

Figure 1A shows the effects of different concentrations of PM on cell viability. The results demonstrated that 100, 200, 300, and 400 µg/mL PM decreased A549 cell viability to 71.6 ± 0.8, 68.1 ± 1.3, 65.0 ± 1.5, and 56.6 ± 1.1%, respectively. Figure 1B shows the effects of WECF on A549 cell viability. We found no significant differences in cell viability with WECF treatment ranging from 10 to 300 µg/mL compared to the control, indicating that WECF in this range did not show any cytotoxic effects toward the viability of A549 cells. Therefore, a dose of 400 µg/mL PM and WECF concentrations of less than 300 µg/mL were selected in subsequent experiments. Figure 1C shows that the cell viability of A549 cells induced by 400 µg/mL PM in the absence of WECF was 65.69%, which indicates that 400 µg/mL shows marked cytotoxicity to A549 cells. Treatment with WECF at 50–300 µg/mL and the addition of 400 µg/mL PM resulted in dose-dependent increases in cell viability compared to the cells treated with 400 µg/mL PM alone. Compared to the untreated cells, PM significantly increased the release of LDH to the culture medium. However, it was reversed by WECF with administration at 50–300 µg/mL (Figure 1D). Clearly, PM exerts its cytotoxicity in A549 cells, whereas WECF significantly protects A549 cells from PM-induced damage and significantly reduces LDH release in A549 cells. 

### 2.2. Effects of WECF on Antioxidant Indices

To evaluate the effect of WECF on PM-induced A549 cell oxidation, the enzymatic and nonenzymatic antioxidant indices in A549 cells were determined. As shown in Figure 2, ROS generation (Figure 2A) and TBARS formation (Figure 2B) decreased in the cells treated with WECF. GSH contents (Figure 2C) and SOD activity (Figure 2D) increased in cells treated with WECF compared to the cells treated with PM alone. The results obtained from Figure 2 show that PM significantly increased oxidative stress in A549 cells as compared to the untreated cells. WECF suppressed PM-induced oxidative stress, in a dose dependent manner, compared to the cells treated with 400 µg/mL PM alone. These observations indicate that WECF can attenuate oxidative stress in PM-induced cells due to its antioxidant potential.

### 2.3. Effect of WECF on Nuclear Factor-Erythroid 2-Related Factor 2 (Nrf2) and Heme Oxygenase-1 (HO-1) Protein Expression

Figure 3 shows the effects of WECF on Nrf2 and HO-1 protein expression in PM-induced A549 cells. As shown in Figure 3A,B, 400 µg/mL PM decreased Nrf2 and HO-1 expression to 0.81-fold and 0.54-fold, compared to the untreated cells, respectively. However, treatment with 50, 150, and 300 µg/mL WECF increased the expression of Nrf2 by 1.18-, 171-, and 2.42-fold, respectively, and the expression of HO-1 by 0.57-, 0.78-, and 0.86-fold, respectively, compared to the untreated cells. Obviously, WECF upregulates Nrf2 and HO-1 expression in PM-induced A549 cells. That is to say, WECF can protect PM-induced A549 cells from oxidative stress through the activation of the Nrf2 pathway.

### 2.4. WECF Regulation of Nrf2 and HO-1 via the Mitrogen-Activated Protein Kinase (MAPK) Signaling Pathway

In order to evaluate the molecular mechanism of regulation of the Nrf2 and HO-1 proteins, cotreatment of the MAPK signaling pathways with a p38 inhibitor (SB203580), a JNK inhibitor (SP600125), and an ERK inhibitor (U0126) using Western blotting was examined. As shown in Figure 3C, the Nrf2 expression was 0.84-fold in PM-induced cells compared to the control; however, after treatment with WECF at 300 µg/mL without PM, Nrf2 expression was 1.23-fold in WECF-treated cells, compared to the untreated cells. When cells were cotreated with PM and WECF, Nrf2 expression was 1.16-fold compared to the untreated cells. In cells treated with PM, WECF, and a JNK inhibitor (SP600125) or an ERK inhibitor (U0126), Nrf2 expression was 0.67- and 0.45-fold, compared to the untreated cells, respectively. The expression of Nrf2 was significantly decreased in both SP600125- and U0126-treated cells, compared to the cells treated with WECF plus PM. There was no significant difference between SB203580-treated cells and the cells treated with WECF plus PM group. These results indicate that WECF regulates Nrf2 expression through the JNK and ERK pathways. As shown in Figure 3D, PM significantly decreased HO-1 activity (0.74-fold) as compared to the untreated cells; however, it was reversed by the addition of WECF (0.86-fold), indicating that WECF positively regulates HO-1 expression, but there was no significant difference between PM-induced cells with WECF and the cells treated with PM alone. In addition, the expression of HO-1 was significantly reduced in both SP600125- and U0126-treated cells, compared to the cells treated with WECF plus PM. The HO-1 expression was reduced by SB203580-treated cells, but no significant difference was found between SB + WECF + PM-treated cells and PM + WECF-treated cells. These results show that treatment with SP600125 (a JNK inhibitor) and U0126 (an ERK inhibitor), but not SB203580 (a p38 MAPK inhibitor), reduced WECF-induced Nrf2 and HO-1 upregulation in A549 cells exposed to PM, indicating that WECF regulates Nrf2 and HO-1 expression through the JNK and ERK pathways. These results imply that WECF protected A549 cells from PM-induced oxidative damage by increasing Nrf2 and HO-1 expression via the ERK and JNK signaling pathways.

### 2.5. Bioactive Compounds in WECF

Some bioactive compounds in plants can show relevant biological effects in reducing the risk of different human diseases. Therefore, it is necessary to analyze and identify the bioactive compounds in WECF. Figure 4 shows the HPLC-MS total ion and HPLC-DAD chromatogram for WECF. Table 1 summarizes the retention times, UV-VIS wave length maxima (λmax), (+)ESI-MS masses [M+H]^+^, other ion fragments from (+)ESI-MS, (-)ESI-MS masses [M-H]^−^, MS/MS product ions of the [M-H]^−^, and the content of the fourteen compounds of WECF. Structure identification of these compounds in the water extracts of djulis was elucidated by comparison with the product ion spectra of known compounds, which have been reported from previous studies. Of these fourteen compounds, seven compounds are phenolic compounds. In addition to the presence of various phenolic compounds, amaranthin, iso-amaranthin, betanin, and isobetanin represent the pigments which account for the red colors of djulis. Among the fourteen identified compounds, the total contents of quercetin derivative (8), quercetin-3-*O*-rutinoside-7-*O*-rhamnoside (9), quercetin-3-*O*-triasccharide (10), and qucecetin-3-*O*-(2, 6-di-*O*-rhamnosyl-glucoside (11) account for 2648.2 µg/g, while rutin accounts for 2219.7 µg/g. In addition, flavonoid glycosides can be metabolized to aglycones by the colon microflora. Moreover, in our previous works, betanin and kaempferol present in WECF demonstrated significant protective effect against oxidative damage *in vitro* and *in vivo* [12,15]. Considering that quercetin glycosides and rutin are the two most abundant components in WECF as quantified by using the peak of HPLC-DAD analysis (Table 1) and that betanin and kaempferol have biological effects, quercetin, rutin, betanin, and kaempferol were selected for determination of their protective effects on PM-induced A549 cell cytotoxicity. The cytotoxicity of rutin, quercetin, betanin, and kaempferol in A549 cells was determined using 3-(4,5-dimethylthiazol-2-yl)-2,5-diphenyltetrazolium bromide (MTT) assay. Rutin, quercetin, betanin, and kaempferol did not show any cytotoxic effects on the A549 cell growth at 5 µM (data not shown). To understand if rutin, quercetin, betanin, and kaempferol are responsible for protecting A549 cells from oxidative damage induced by PM, the effects of rutin, quercetin, betanin, and kaempferol on PM-induced A549 cell growth were determined. As shown in Figure 5A, treatment of A549 cells with rutin and quercetin at 1 and 5 µM significantly protected A549 cells from PM-induced cytotoxicity. However, no significant differences were found between the PM-induced cells treated with betanin and kaempferol and the cells treated with PM alone. Obviously, rutin and quercetin can alleviate the cell damage induced by PM in A549 cells. Therefore, rutin and quercetin were selected as reference compounds for subsequent experiments to determine their protective effects against PM-induced oxidative damage in A549 cells. As expected, the treatment of A549 cells with rutin and quercetin at 1 and 5 µM significantly protected A549 cells from PM-induced ROS generation (Figure 5B) and TBARS formation (Figure 5C). In addition, the exposure of PM-induced A549 cells to rutin and quercetin at 1 and 5 µM resulted in increased levels of GSH (Figure 5D) and SOD activity (Figure 5E) compared to the cells treated with PM alone. Moreover, rutin at 1 and 5 µM significantly increased Nrf2 expression in the PM-induced A549 cells (Figure 5F); however, quercetin did not affect Nrf2 expression. HO-1 expression was increased by rutin at 1 and 5 µM in PM-induced cells, but no significant difference was found between the PM-induced cells treated with rutin and the cells treated with PM alone (Figure 5G). For the PM-induced cells treated with quercetin at 1 and 5 µM, no significant difference was found in HO-1 expression between the PM-induced cells treated with quercetin and the cells treated with PM alone (Figure 5G). These findings imply that quercetin at 1 and 5 µM had no significant effect on Nrf2 and HO-1 protein expression in PM-induced A549 cells.

## 3. Discussion

A PM review by Feng et al. [6] indicated that PM2.5 can cause airway inflammation, a decline in lung function, the incidence and exacerbation of asthma and chronic obstructive pulmonary disease (COPD), and render the lungs susceptible to infections. Given that the lungs are in direct contact with atmospheric air pollution, A549 cells, a human lung cell line with the characteristic features of the type II cells of the pulmonary epithelium [18], were used as an *in vitro* model for evaluating the cytoprotective potential of djulis and its bioactive compounds. In addition, considering that ambient air pollution has caused millions of annual premature deaths globally [1], an ambient urban dust PM sample, standard reference material (SRM) 1649b, which has recently been widely used as the reference PM sample in many studies [19], was selected as the source of urban dust material in the present study.

Many studies have provided evidence that natural sources could potentially decrease PM-induced diseases [20]. In our previous studies, WECF showed marked biological activities against the cytotoxicity induced by oxidative stress [15]. Therefore, this study further explores whether WECF may protect cells against PM-induced oxidative damage. LDH is widely used as a marker to study the toxicity of toxicants [15]. Figure 1D shows that A549 cells treated with PM show a significant release of LDH after 24 h. However, a significant decrease in the LDH released from cells was observed after exposure to WECF, indicating that WECF prevented PM-induced cell death. In addition, according to the results from Figure 2A, PM induced ROS generation (Figure 2A) and TBARS formation (Figure 2B), indicating that A549 cells exposed to PM can trigger oxidative stress, thereby leading to various adverse effects on cells. Interestingly, WECF suppressed the PM-induced oxidative stress due to repression of ROS generation and TBARS formation. Moreover, the reduction of ROS generation (Figure 2A) and TBARS formation (Figure 2B) parallel the cytoprotective effects of PM-induced A549 cells (Figure 1C,D). In other words, the inhibition of oxidative damage by treatment with WECF may contribute to protecting the A549 cells from PM-induced oxidative damage. Moreover, WECF in the range from 10–300 µg/mL did not have any cytotoxic effects toward A549 cell growth, indicating that WECF does not produce apoptotic activity in A549 cells. A growing body of evidence from *in vitro* and *in vivo* experiments has indicated that PM exposure may cause systemic oxidative damages in living tissues [21]. The possible modes of action may include: (1) There are environmentally persistent free radicals in PM. (2) Numerous organic chemicals coated on PM can be metabolically activated into PM, which may produce or increase intracellular ROS. (3) Transition metals, such as Fe, Cu, and Mn, present in PM may also induce ROS via the Fenton reaction. (4). The SRM 1649b used as PM in this study is composed of polycyclic aromatic hydrocarbons (PAHs), nitro-PAHs, polycyclic biphenyls (PCB), chlorinated pesticides, decabromodiphenyl ether, dioxin, nitrates, sulfates, and metals such as nickel, copper, chromium, manganese, vanadium, and aluminum [19,22]. These compounds, present in PM, may induce oxidative damage in A549 cells. In our previous studies, WECF showed significant antioxidant activity [15]. In addition, the results from Figure 2A,B indicate that WECF enhances the antioxidant indices in PM-induced A549 cells. These observations probably explain that the cytoprotective effect of WECF on PM-induced cell death may in part result from the inhibition of oxidative stress in A549 cells. Non-enzymatic antioxidants, glutathione, and antioxidant enzymes, such as superoxide dismutase (SOD), catalase (CAT), and glutathione peroxidase (GSHPX), are the main antioxidant-defense mechanisms in mammals [10]. This defensive system, including antioxidant enzymes and antioxidants, deoxidizes ROS and reduces oxidative damage to cells. However, it is well-known that PM can impair the antioxidant system and reduce the antioxidant potential of the exposed cells [6]. In this study, PM-induced ROS generation and lipid peroxidation in A549 cells was attenuated by WECF. A reasonable explanation is that the observed cytoprotective potential of WECF may reflect mainly their direct actions on mediators of PM toxicity, leading to the reduction of ROS generation and lipid peroxidation [23]. SOD is a well-known oxyradical detoxification enzyme in living cells exposed to oxygen. As shown in Figure 2D, the enzyme activity of SOD decreased significantly in PM-induced cells while WECF pretreatment significantly increased, in a dose dependent manner, the SOD activity when compared to the cells treated with PM alone. This result suggests that SOD activity, enhanced by WECF, plays an important role in PM-induced A549 cells, which reinforces the cells to prevent PM-induced oxidative damage. Furthermore, since GSH is an important intracellular antioxidant and reducing agent and plays an important role as a co-factor in the detoxification of oxidants and toxic xenobiotics, GSH is widely used as a biomarker of the redox state in intracellular cells. For this reason, it is necessary to determine whether WECF is able to upregulate the non-enzymatic components in PM-induced cells. As shown in Figure 2C, WECF in the range from 50 to 300 µg/mL produces a remarkable increase in GSH levels compared to the PM-induced cells. This result suggests that WECF maintains the normal redox status of A549 cells and suppresses the oxidative damage induced by PM. In this regard, the regulation of the contents of GSH by WECF may be considered as an effective approach to preventing PM-induced oxidative stress in cells. Malondialdehyde, one of the products formed through the decomposition of lipid peroxidation, is widely recognized as a biomarker of the lipid peroxidation caused by oxidative stress [24]. In this study, a statistically significant increase in cells treated with PM was observed (Figure 2B), indicating that a significant degree of lipid peroxidation in PM-induced cells occurred. These observations indicate that the degree of membrane lipid peroxidation in PM-induced cells increased, causing MDA accumulation, which may make the cell contract, thereby damaging membranes and the cells. However, for the PM-induced cells treated with WECF, the degree of lipid peroxidation was significantly repressed. Based on the results from Figure 2A,B, WECF reduced the negative effects of ROS and lipid peroxidation of A549 cells induced by PM. This finding is in accordance with the determination of SOD activity and GSH content, which indicated that the increase in GSH levels and SOD activity reduced the ROS generation and MDA formation. GSH is synthesized by consecutive reactions of two enzymes, γ-glutamylcysteine (γGluCys) synthetase and GSH synthetase. In addition, phytochemicals may stimulate the synthesis of antioxidant enzymes and detoxification systems at the transcriptional level and may increase glutamylcystein synthesis through the antioxidant response system [25]. This observation seems to be in line with the protection against PM-induced A549 cell death. In other words, the maintenance of cellular homeostasis in PM-induced A549 cells is associated with nonenzymatic components, GSH, and the antioxidant enzyme SOD, which is enhanced by treatment with WECF and ultimately maintains the redox balance of the intracellular environment [26].

Many polyphenolic compounds exert healthy and biofunctional effects through a major pathway, which includes the induction of intracellular oxidative stress, partly mediated by the aryl hydrocarbon receptor (AHR), and AHR activation promotes Nrf2 activity, thereby leading to an intracellular defense response [27]. In normal physiological conditions, Nrf2 is fixed in the cytoplasm by the cytoskeletal protein kelch-like ECH associated protein 1 (keap1). Once Nrf2 is under oxidative stress, it can exert transcriptional activity after a transmembrane transfer into the nucleus and binding to the antioxidant response element (ARE) promoter sequence or the DNA binding sequence, which subsequently induce the expression of downstream target genes, such as SOD and HO-1 enzymes [28]. According to the results, PM-exposure was shown to alter the expression of HO-1 (Figure 3B) and decrease SOD activity (Figure 2D) in PM-induced cells, indicating that PM can impair the antioxidant enzymes and decrease the defense system of the exposed cells. However, WECF treatment can enhance the levels of HO-1 (Figure 3B), increase the SOD activity (Figure 2D), and promote the transcription of Nrf2 (Figure 3A) into the nucleus in a concentration-dependent manner. Thus, the expression of HO-1 is positively correlated with Nrf2 protein expression, indicating that WECF can upregulate HO-1 activity through Nrf2 signaling in PM-induced A549 cells [28]. This finding may explain that the inhibitory effect of WECF on the oxidative stress induced in the cells by PM is, in part, attributable to the upregulation of HO-1 activity via Nrf2 signaling.

MAPK, which is an important mediator of cell membrane to nucleus signal transduction in response to oxidative stress, plays a central role in cell growth, survival, differentiation, and apoptosis [29]. Therefore, the role of MAPK in WECF-treated cells induced by PM was further examined. To evaluate the role of MAPK in the signaling pathway and to understand which MAPK plays a crucial role in regulating PM-induced oxidative damage in A549 cells treated with WECF, the cells were treated with the MAPK-specific inhibitors SP600125 (JNK inhibitor), U0126 (ERK inhibitor), and SB203580 (p38 inhibitor). As shown in Figure 3C,D, the expression of Nrf2 and HO-1 in PM-induced cells treated with WECF was significantly inhibited by the JNK an ERK inhibitors, while the p38 inhibitor had no significant effect. These results imply that the transcription factor Nrf2 is up-regulated through the ERK and JNK signaling pathways in WECF-treated A549 cells induced by PM. Obviously, WECF protected A549 cells from PM-induced oxidative damage by upregulating the expression of Nrf2 and HO-1 via the ERK and JNK signaling pathways.

The accumulating evidence shows that bioactive compounds in plants may play an important role in the maintenance of a healthy lifestyle. These bioactive compounds are ubiquitous in the plant kingdom and are considered as non-nutritional but vital ingredients for preventing living cells from cytotoxic damage. In the present study, rutin, quercetin, and twelve other compounds were identified as present in WECF. To compare with our previous report [15], rutin and quercetin derivatives were found as the major compounds in the study, which was slightly different from the previous report that showed that rutin, quercetin, and kaempferol derivatives were the major compounds, revealing that the number of identified compounds may vary each time. The differences between the two results may be due to the differences in planting, harvesting, storage conditions, and climate changes [12]. Recently, several bioactive compounds have been thoroughly investigated for their human health-maintaining biological effects using various *in vitro* and *in vivo* models, as well as clinical trials [30]. Therefore, the effects of bioactive compounds on PM-induced oxidative damage of A549 cells were further explored. As expected, rutin and quercetin significantly reduced ROS generation (Figure 5B) and TBARS formation (Figure 5C) and enhanced GSH levels (Figure 5D) and antioxidant enzyme (SOD) (Figure 5E) while also up-regulating Nrf2 protein expression in PM-induced A549 cells (Figure 5F). However, quercetin at 1 and 5 µM had no effect on Nrf2 (Figure 5F) and HO-1 (Figure 5G) protein expression in PM-induced A549 cells. It has been suggested that bioactive compounds, such as quercetin, may act as a ‘double-edged sword’ due to its unique properties, since it behaves as an antioxidant and/or pro-oxidant, depending on its concentration and the duration of exposure [31]. Therefore, in the present work, we speculate that quercetin at 1 and 5 µM co-cultured with PM-induced cells for 12 and 15 h might be not the appropriate time or concentration to positively regulate the protein expression of Nrf2 and HO-1, respectively. However, this speculation requires further study. Obviously, these results imply that the cytoprotective effect of WECF on PM-induced *in vitro* cell death may, in part, be attributable to the antioxidant activity of A549 cells treated with rutin and quercetin. However, betanin and kaempferol did not show significant protective effects against PM-induced A549 cell growth. Moreover, the twelve compounds identified as present in WECF may contribute to cytoprotection against PM-induced oxidative damage, although the antioxidant potentials of the compounds were not determined. However, these compounds, such as betanin [11,12,15,17,32], kaempferol [33], and protecatechuic acid [34], have been shown to stimulate biofunctional effects on living cells [14]. In addition, rutin, quercetin, kaempferol, and protecatechuic acid are known as compounds with a phenolic group in their structure, and therefore they may act as antioxidants, since they donate electrons and undergo electron delocalization [35]. In other words, the higher the number of bioactive compounds with phenolic structures in WECF, the higher the biological activity. According to the data from Table 1 and Figure 4, there are other unidentified compounds in WECF, along with the bioactive compounds identified. Moreover, there are many complex interactions, including additive, synergistic, and/or antagonistic effects, depending on the structure and condition, between the compounds present in WECF [14]. We can infer from the data that the inhibitory effect of WECF on PM-induced oxidative damage in A549 cells may rely partly on these bioactive compounds due to a direct action or the synergy between the compounds, thereby leading to the alleviation of PM-induced oxidative damage in A549 cells.

## 4. Materials and Methods

### 4.1. Chemicals and Reagents

MTT was purchased from Sigma-Aldrich Chemical Co. (St. Louis, MO, USA). The MAPK pathway inhibitors, including p38 inhibitor (SB203580, SB), JNK inhibitor (SP600125, SP), and ERK inhibitor (U0126, U), were obtained from Cayman Chemical (Ann Arbor, MI, USA). The Nrf2 antibody and HO-1 antibody were purchased from Cell Signaling Technology (Danvers, MA, USA). Human lung adenocarcinoma A549 cells (BCRC number: 60074) were purchased from the Bioresource Collection and Research Center (BCRC, Food Industry Research and Development Institute, Hsinchu, Taiwan). The chemicals used in the research were analytical grade.

### 4.2. WECF Preparation

The dehusked djulis (*Chenopodium formosanum*) was obtained from Colaidea Co., Ltd., Pingtung, Taiwan. Dehusked djulis, identified by Professor Yau-Lun Kuo of the Department of Forestry, National Pingtung University of Science and Technology, Pingtung, Taiwan, was purchased from Kullku Farm (Pingtung, Taiwan) and was ground to a fine powder with a high-speed grinder (RT08, Rong Tsong, Taipei, Taiwan) before the extraction. The voucher specimen (No. CNU-101) was deposited in the Department of Food Science and Technology, Chia Nan University of Pharmacy and Science, Tainan, Taiwan [13]. Dehusked djulis was extracted with boiling water at the ratio of 1:10 (*w*/*v*) for 20 min. After filtering through Advantec No. 2 filter paper (Toyo Roshi Kaisha, Ltd., Tokyo, Japan), the residue was re-extracted under the same conditions. All of the supernatants were combined and concentrated by a rotary evaporator under reduced pressure and freeze-drying. The water extracts of djulis, abbreviated as WECF, were stored at −20 °C until they were used.

### 4.3. PM Preparation

The particulate matter (PM), urban dust standard reference material (Standard Reference Material, SRM 1649b Urban Dust), was purchased from the National Institute of Standards and Technology (NIST, Gaithersburg, MD, USA). Fifty milligrams of PM were suspended in 1 mL of dimethyl sulfoxide and sonicated for 120 min. The PM suspension was then centrifuged at 13,000× *g* for 30 min, and any impurities were removed using a 0.22 µm syringe filter and stored at −20 °C for use [20].

### 4.4. HPLC/ESI-MS-MS Analysis of WECF

For removing the water-soluble impurities, i.e., sugars, colorants, etc., solid phase extraction was applied for the sample treatment of freeze-dried powder from aqueous extract of djulis, as previously described [36]. In brief, a sample of djulis freeze-dried powder (1.0 g) was diluted with water (10 mL, containing 100 µg of internal standard: 7-methoxyflavanone, Sigma) and centrifuged at 13,000× *g* for 30 min to remove insoluble matters. The supernatant was applied to a pre-conditioned (washed with methanol and then water) Oasis HLB column from Waters (60 mg/3 mL, Milford, MA, USA). After loading, the column was washed with water (5 mL) and then eluted with a solution of 30% methanol in water (5 mL). An aliquot of 5 mL of the eluates was dried in a vacuum concentrator and redissolved in 1 mL of methanol before applying it on the HPLC-MS analysis.

The HPLC/electrospray ionization (ESI) mass spectrometric analysis of aqueous extract of djulis was conducted according to previous a report [12] with minor modifications. In brief, the analysis of the prepared extracts was performed using a Waters HSS T3 (2.1 × 150 mm, 1.8 µm, Waters Corp., Milford, MA, USA) analysis column fitted with a Security-Guard Ultra C18 guard column (2.1 mm × 2.0 mm, sub-2 µm, Phenomenex, Inc., Torrance, CA, USA) using an HPLC system with a photodiode-array (PDA) detector. The elution solvent system was performed by gradient elution using two solvents: Solvent A (water containing 0.1% formic acid) and Solvent B (acetonitrile containing 0.1% formic acid). The flow rate during the elution process was 0.2 mL/min and the column temperature was set at 35 °C. The binary gradient elution was conducted as follows: 0–3 min (2% B), 3–6 min (2–10% B), 6–25 min (10–75% B), 25–30 min (75–95% B), 30–40 min (95% B), and 40–45 min (95–2% B). The absorption spectra of the eluted compounds were scanned within 210 to 600 nm using the in-line PDA detector monitored at 254, 280, 360, and 530 nm, respectively. The compounds, having been eluted and separated, were further identified with a triple quadruple mass spectrometer. The system was operated in electrospray ionization (ESI) with both positive and negative ionization modes in a potential of + and −3500 V, respectively, applied to the tip of the capillary. Samples of 10 µL of extracts were directly injected into the column using an autosampler. Nitrogen was used as the drying gas at a flow rate of 10 L/min and the nebulizing gas at a pressure of 30 psi. The drying gas temperature was maintained at 325 °C. The fragmentor voltage was 115 V, and the in-source collision-induced dissociation (CID) voltage was 15 V. Nitrogen was also used as a collision gas. Quadrupole 1 filtered the calculated *m*/*z* of each compound of interest while quadrupole 2 scanned for ions produced by nitrogen collision between these ionized compounds in the range of 100–800 amu at a scan time of 200 ms/cycle. The identification of separated compounds was carried out by comparing their mass spectra provided by ESI-MS and ESI-MS/MS with those of authentic standards when available. Quantitative measurement of the level of each compound in the aqueous extract of djulis was achieved by an internal standard approach with 7-methoxyflavanone.

### 4.5. Cell Culture and Cell Viability Assay

A549 cells, cultured in 90% Ham’s F12K medium, were supplemented with 10% (*v*/*v*) fetal bovine serum and 2 mM L-glutamine at 37 °C in a humidified atmosphere containing 5% CO_2_. The viability of A549 cells was determined by MTT assay. Cells were seeded in 96-well plates at a density of 2 × 10^4^ cells/well and incubated for 24 h. After the cells were treated with WECF at 10–500 µg/mL and 1 and 5 μM bioactive compound for 30 min, PM at 100–400 µg/mL was added to the medium and incubated at 37 °C for 24 h. Then, 50 µL of 0.1% MTT was added in each well and incubated at 37 °C for 1 h. Subsequently, the medium was removed, l00 µL of dimethyl sulfoxide (DMSO) was added to dissolve the colored formazan crystal and measured at 540 nm using an ELISA reader (Molecular Devices VMax, Visalia, CA, USA) [37].

### 4.6. Determination of Cell Leakage Rate

The cell leakage rate was tested using lactate dehydrogenase (LDH) Cytotoxicity Colorimetric Assay Kit II (BioVision Inc., Milpitas, CA, USA). A549 cells were seeded in a 24-well microplate at the density of 1 × 10^5^ cells/mL. After 24 h incubation, the cells were treated with WECF (50–300 µg/mL) for 30 min, and 400 µg/mL PM was added to the medium and incubated for 24 h. After incubation, 10 µL of extracellular fluid from each of the 24 wells was transferred to a 96-well microplate, and 100 µL of LDH reagent was added and incubated for 30 min in the dark at room temperature. The optical density of each well was determined at 450 nm using an enzyme-linked immunosorbent assay (ELISA) reader (Molecular Devices). The LDH leakage was evaluated from the ratio between the enzymatic activity of LDH in the medium and that of the whole cell content [38].

### 4.7. Determination of Intracellular Reactive Oxygen Species

The probe used to detect intracellular reactive oxygen species (ROS) was 2′,7′-dichlorofluorescin diacetate (DCFH-DA). A549 cells were seeded in 6-well plates at a density of 4 × 10^5^ cells/well and incubated for 24 h. After the cells were treated with WECF (50–300 µg/mL) and 1 and 5 μM bioactive compound for 30 min, 400 µg/mL PM was added to the medium and incubated at 37 °C for 2 h. After incubation, the fluorescence intensity was measured by a Bio-Tek FLx 800 microplate fluorescence reader (excitation wavelength: 485 nm; emission wavelength: 528 nm) and the ROS produced from intracellular stress was expressed as the percentage of fluorescence intensity relative to the negative control [39].

### 4.8. Determination of Intracellular Lipid Peroxidation

Malondialdehyde (MDA) is the final product and used biomarkers for lipid peroxidation. Measurement of MDA was based on a reaction with thiobarbituric acid (TBA) to generate a product, thiobarbituric acid-reactive substances (TBARS), which could be detected fluorometrically. A549 cells were seeded in 6-well plates at a density of 4 × 10^5^ cells/well and incubated for 24 h. After the cells were treated with WECF (50–300 µg/mL) and 1 and 5 μM bioactive compound for 30 min, 400 µg/mL PM was added to the medium and incubated at 37 °C for 2 h. After incubation, 1 mM butylated hydroxyanisole (BHA) was added, the reaction solution was taken out and mixed with trichloroacetic acid (TCA) and TBA for 10 min in boiling water. The fluorescence intensity was measured by a Bio-Tek FLx 800 microplate fluorescence reader (excitation wavelength: 530 nm; emission wavelength: 560 nm) [40].

### 4.9. Determination of Intracellular Glutathioine (GSH)

5-chloromethylfluorescein diacetate (CMF-DA) is a fluorescent dye that freely passes through cell membranes into cells for the detection of glutathione. A549 cells were seeded in 6-well plates at a density of 4 × 10^5^ cells/well and incubated for 24 h. After the cells were treated with WECF (50–300 µg/mL) and 1 and 5 μM bioactive compound for 30 min, 400 µg/mL PM was added to the medium and incubated at 37 °C for 20 h. After incubation, cells were washed with phosphate buffered saline (PBS) and treated with 5 μM CMF-DA for 30 min. Then, cells were washed with PBS and intracellular GSH was determined by a Bio-Tek FLx 800 microplate fluorescence reader (excitation wavelength: 485 nm; emission wavelength: 528 nm) [41].

### 4.10. Evaluation of Superoxide Dismutase (SOD)

SOD activity was measured by the Sigma-Aldrich SOD assay kit (St. Louis, MO, USA) according to the manufacturer’s protocol. The sample and each reagent (No. 1, 2, 3, and 4) were thoroughly mixed and measured after 20 min at 37 °C. The optical density was measured at 450 nm in a microplate spectrophotometer (Thermo Multiskan GO, Waltham, MA, USA).

### 4.11. Western Blot Analysis

Western blots were performed as previously described [15] with some modifications. In brief, A549 cells were seeded on a 6 cm dish (1 × 10^6^ cells/dish) and cultured for 24 h. After the cells were treated with WECF (50–300 µg/mL) and 1 and 5 μM bioactive compound for 30 min, 400 µg/mL PM was added to the medium and incubated at 37 °C for 12 and 15 h to determine the expression of Nrf2 and HO-1, respectively. Following culture, cells were harvested and lysed in the ice-cold lysis buffer and kept on ice for 30 min. The obtained cell lysates were quantified by the bicinchoninic acid (BCA) method (Pierce, Rockfold, IL, USA). Each sample, which contained 100 μg of proteins, was separated using 10% sodium dodecyl sulfate polyacrylamide gel electrophoresis (SDS-PAGE) and transferred to a nitrocellulose (NC) membrane (Pall, New York, NY, USA). The nonspecific binding sites of the membrane were blocked with 5% BSA in PBST (0.1% *v*/*v* Tween-20 in PBS, pH 7.2) for 1 h and the membranes were washed with PBST buffer three times. Then, they were immunoblotted with anti-Nrf2 (#12721S) and anti-HO-1 (#5853S) against Nrf2 (1:1000), HO-1 (1:1000) (Cell Signaling Technology) overnight at 4 °C. After washing with PBST, the membranes were incubated with horse radish peroxidase-conjugated secondary antibody (Santa Cruz, Dallas, TX, USA) for 1 h at room temperature. Then, the membranes were washed with PBST buffer three times. After visualization was conducted using an Enhanced Chemiluminescence (ECL) plus kit (Amersham Bioscience, Aylesbury, UK) and imaging was conducted using an ECL detection system. β-actin (45 kDa protein) was used as an internal standard. The corresponding bands to determine the strength of protein expression were analyzed by Image J software (version 1.52 v, National Institutes of Health, Bethesda, MD, USA). The expression levels of Nrf2, HO-1, and β-actin proteins were determined by densitometry and analyzed.

### 4.12. MAPK Inhibitors Assay

To explore signaling pathways, the kinase inhibitors [U0126 for the mitogen-activated protein kinase MAPK ERK, SB203580 for p38 MAPK, and SP600125 for c-Jun N-terminal kinase (JNK)] (Cayman Chemical, Ann Arbor, MI, USA) were used. Each inhibitor (20 µM in all cases) was added to the cell culture medium for 1 h before WECF. After the cells were treated with WECF (300 µg/mL) for 1 h, 400 µg/mL PM was added to the medium and incubated at 37 °C for 12 and 15 h for determination of Nrf2 and HO-1 protein expression, respectively. The determination of Nrf2 and HO-1 protein expression was conducted by Western blot analysis, as described above.

### 4.13. Statistical Analysis

Each experiment was performed at least in triplicate and the results were averaged. Data are expressed as means ± SD, and ANOVAs were conducted by using the SPSS software (version 12.0, SPSS Inc., Chicago, IL, USA). When a significant F ratio was obtained (*p* < 0.05), a post hoc analysis was conducted between the groups by using a Duncan’s multiple range test. A significant difference between treatments was considered when *p*-values were less than 0.05.

## 5. Conclusions

Based on the study results, WECF showed a marked protective effect against PM-induced oxidative damage in A549 cells through the increase of GSH content, SOD activity, the expression of antioxidant enzymes, and the expression of Nrf2. In addition, protein expression of Nrf2 and HO-1 were significantly reduced after treatment with SP600125 (a JNK inhibitor) and U0126 (an ERK inhibitor), further confirming that WECF-treated protection is regulated by the JNK and ERK signaling pathways. This observation supports the modulating effect of WECF on Nrf2. In other words, WECF induced endogenous antioxidant defense mechanisms by modulating transcription factors, such as Nrf2. Moreover, the identification of bioactive compounds, such as rutin and quercetin, against oxidative damage induced by PM may help to understand their protective association with oxidative stress induced by PM. This study suggests that WECF has the potential for use in functional foods and in preventing PM-induced oxidative damage in living cells and tissues. However, further in-depth *in vivo* study is required to verify this speculation.

## Figures and Tables

**Figure 1 molecules-27-00253-f001:**
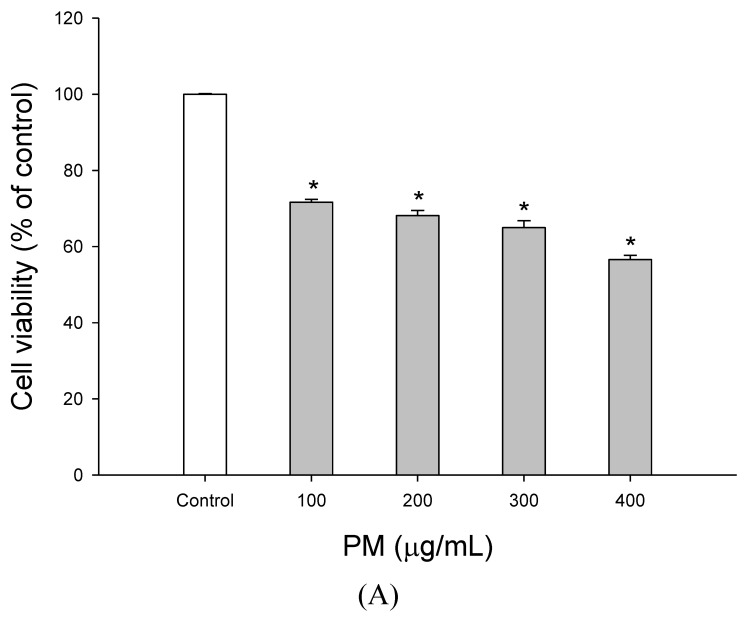
Effects of water extracts of djulis (WECF) on the A549 cell viability induced by particulate matter (PM). (**A**) Effects of different concentrations of PM on A549 cell viability. The cells were treated with PM for 24 h. * (*p* < 0.05) compared with the control group. (**B**) Effects of different concentrations of WECF on A549 cell viability. The cells were treated with WECF for 24 h. * (*p* < 0.05) compared with the control group. (**C**) Effects of WECF on PM-induced A549 cell viability. The cells were treated with WECF and exposed to 400 µg/mL PM for 24 h. ^#^ (*p* < 0.05) compared with the control group and * (*p* < 0.05) compared with 400 µg/mL PM-induced cells alone. (**D**) Effects of WECF on lactate dehydrogenase (LDH) leakage in 400 µg/mL PM-induced A549 cells. The cells were treated with WECF and exposed to 400 µg/mL PM for 24 h. ^#^ (*p* < 0.05) compared with the control group and * (*p* < 0.05) compared with 400 µg/mL PM-induced cells alone (**D**). Data are presented as means ± SD (*n* = 3).

**Figure 2 molecules-27-00253-f002:**
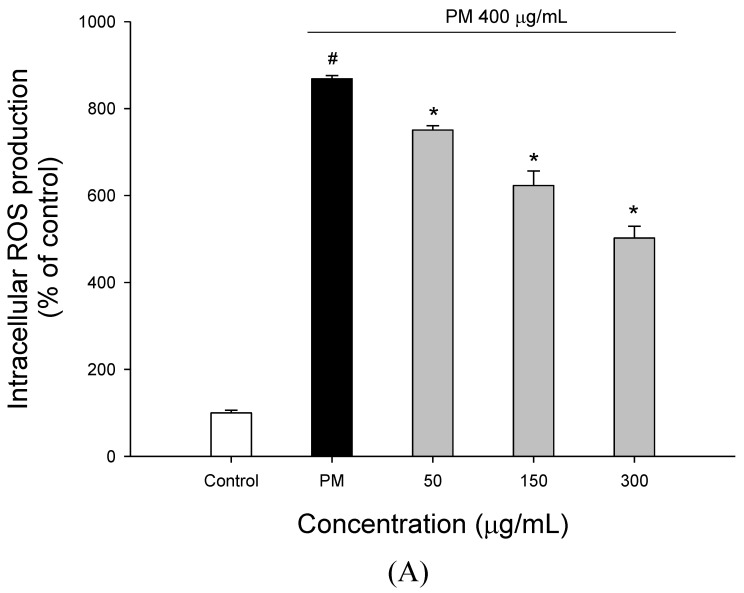
Effects of water extracts of djulis (WECF) on PM-induced oxidation and antioxidant indices in A549 cells. (**A**) Effects of WECF on PM-induced intercellular ROS generation in A549 cells. The cells were treated with WECF and exposed to 400 µg/mL PM for 2 h. (**B**) Effects of WECF on PM-induced intercellular TBARS formation in A549 cells. The cells were treated with WECF and exposed to 400 µg/mL PM for 2 h. (**C**) Effects of WECF on PM-induced glutathione (GSH) contents in A549 cells. The cells were treated with WECF and exposed to 400 µg/mL PM for 20 h. (**D**) Effects of WECF on PM-induced SOD activity in A549 cells. The cells were treated with WECF and exposed to 400 µg/mL PM for 24 h. ^#^ (*p* < 0.05) compared with the control group and * (*p* < 0.05) compared with 400 µg/mL PM-induced cells alone. Data are presented as means ± SD (*n* = 3).

**Figure 3 molecules-27-00253-f003:**
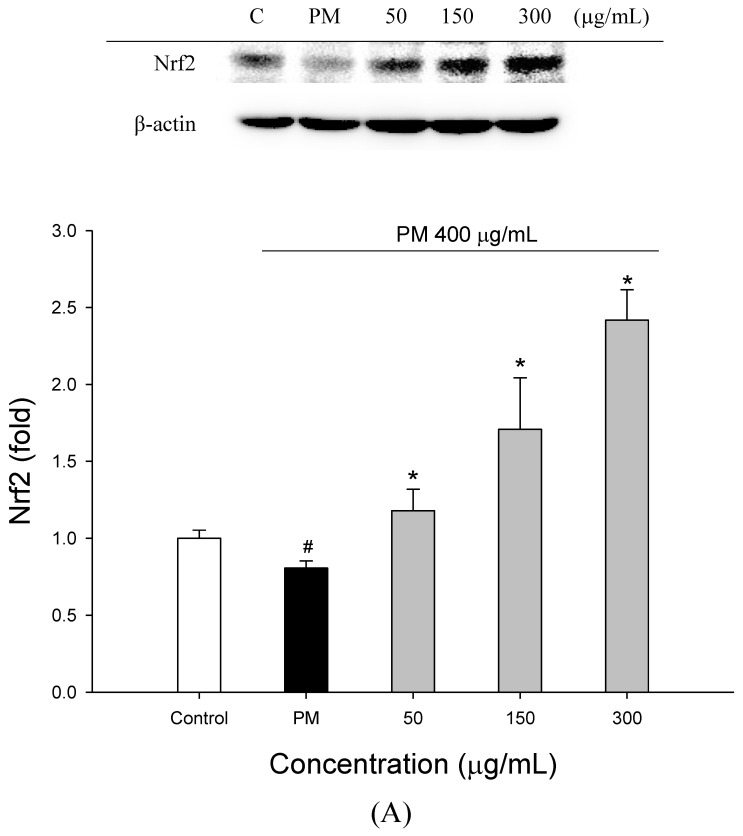
Effects of water extracts of djulis (WECF) on PM-induced Nrf2 and HO-1 protein expression in A549 cells. (**A**) Effects of WECF on PM-induced expression of Nrf2 in A549 cells. The cells were treated with WECF and exposed to 400 µg/mL PM for 12 h. ^#^ (*p* < 0.05) compared with the control group and * (*p* < 0.05) compared with 400 µg/mL PM-induced cells alone. (**B**) Effects of WECF on PM-induced HO-1 activity in A549 cells. The cells were treated with WECF and exposed to 400 µg/mL PM for 15 h. ^#^ (*p* < 0.05) compared with the control group and * (*p* < 0.05) compared with 400 µg/mL PM-induced cells alone. (**C**) Effects of p38 inhibitor (SB203580, SB), JNK inhibitor (SP600125, SP), and ERK inhibitor (U0126, U) on WECF-induced Nrf2 protein expression in PM-treated A549 cells. Control, cultured with medium alone for 12 h; PM, incubated with PM 400 µg/mL for 12 h; P + W, incubated with WECF and PM for 12 h (WECF was added 30 min before PM); SB + W + P, SP + W + P, and U + W + P, treated as described for P + W except that the p38 MAPK inhibitor SB203580, the JNK inhibitor SP600125, or the ERK inhibitor U0126 were added to the medium before WECF. ^#^ (*p* < 0.05) compared with the control group and * (*p* < 0.05) compared with WECF+ 400 µg/mL PM-induced cells. (**D**) Effects of p38 inhibitor (SB203580, SB), JNK inhibitor (SP600125, SP), and ERK inhibitor (U0126, U) on WECF-induced HO-1 protein expression in PM-treated A549 cells. Control, cultured with medium alone for 15 h; PM, incubated with PM 400 µg/mL for 15 h; P + W, incubated with WECF and PM for 15 h (WECF was added 30 min before PM); SB + W + P, SP + W + P, and U + W + P, treated as described for P + W except that the p38 MAPK inhibitor SB203580, the JNK inhibitor SP600125, or the ERK inhibitor U0126 were added to the medium before WECF. ^#^ (*p* < 0.05) compared with the control group and * (*p* < 0.05) compared with WECF+ 400 µg/mL PM-induced cells. Values in each sample with different lowercase letters are significantly different (*p* < 0.05). Data are presented as means ± SD (*n* = 3).

**Figure 4 molecules-27-00253-f004:**
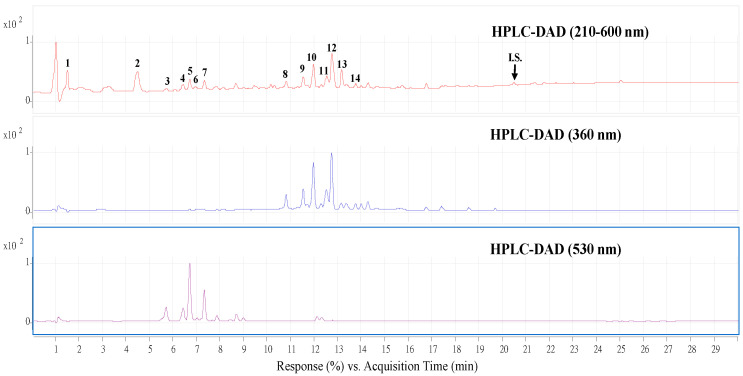
High performance liquid chromatograms detected at a full UV-Vis spectrum of 210–600 nm (**top**), UV 360 (**middle**), and 530 nm (**bottom**) from water extracts of djulis (WECF). Peak numbers refer to Table 1.

**Figure 5 molecules-27-00253-f005:**
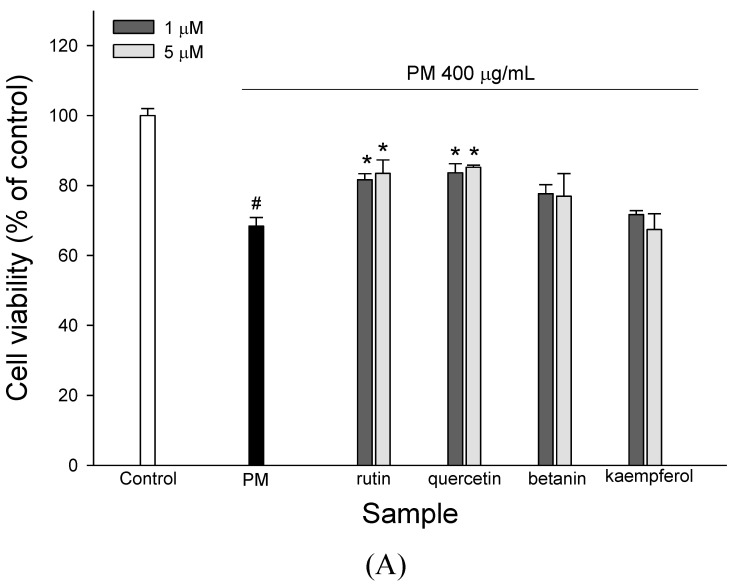
Effects of rutin and quercetin on PM-induced A549 cell viability, oxidation, and antioxidant indices in A549 cells. (**A**) Effects of rutin and quercetin on PM-induced A549 cell viability. The cells were treated with rutin and quercetin, respectively, and exposed to 400 µg/mL PM for 24 h. (**B**) Effects of rutin and quercetin on PM-induced intercellular ROS production in A549 cells. The cells were treated with rutin and quercetin, respectively, and exposed to 400 µg/mL PM for 2 h. (**C**). Effects of rutin and quercetin on PM-induced intercellular TBARS formation in A549 cells. The cells were treated with rutin and quercetin, respectively, and exposed to 400 µg/mL PM for 2 h. (**D**) Effects of rutin and quercetin on PM-induced GSH contents in A549 cells. The cells were treated with rutin and quercetin, respectively, and exposed to 400 µg/mL PM for 20 h. (**E**) Effects of rutin and quercetin on PM-induced SOD activity in A549 cells. The cells were treated with rutin and quercetin, respectively, and exposed to 400 µg/mL PM for 24 h. (**F**) Effects of rutin and quercetin on PM-induced expression of Nrf2 in A549 cells. The cells were treated with rutin and quercetin, respectively, and exposed to 400 µg/mL PM for 12 h. (**G**) Effects of rutin and quercetin on PM-induced HO-1 activity in A549 cells. The cells were treated with rutin and quercetin, respectively, and exposed to 400 µg/mL PM for 15 h. ^#^ (*p* < 0.05) compared with the control group and * (*p* < 0.05) compared with 400 µg/mL PM-induced cells alone. Data are presented as means ± SD (*n* = 3).

**Table 1 molecules-27-00253-t001:** HPLC-DAD-ESI-MS/MS analysis on the chromatographic and spectroscopic characteristics and content of water extract of djulis.

Peak No.	RT (min)	Compound Name	λ_max_ (nm)	[M+H]^+^/ [M−H]^−^, *m*/*z*	MS/MS ^c^ *m*/*z*	Content (µg/g)
1	1.51	Unknown	266	**136** ^b^/134	-	778.8 ± 83.9
2	4.43	Phenylacetic acid derivative *	234, 262	285/**283**	151	1961.1 ± 126.6
3	5.73	Amaranthin	268, 536	**727**/	389	189.9 ± 82.9
4	6.42	Isoamaranthin	264, 530	**727**/	389	411.1 ± 61.8
5	6.70	Betanin ^a^	260, 290sh, 538	**551**/	389	589.4 ± 76.9
6	6.99	Isodopaxanthin	260, 472	**391**/389	255, 150, 345, 347	178.6 ± 48.9
7	7.34	Isobetanin	268, 290sh, 532	**551**/	389	417.0 ± 38.8
8	10.82	Quercetin derivative *	256, 352	889/**887**	741, 446, 300	305.6 ± 49.4
9	11.54	Quercetin-3-*O*-rutinoside-7-*O*- rhamnoside	254, 352	757/**755**	609, 447, 301	473.2 ± 53.6
10	11.94	Quercetin-3-*O*-trisaccharide	228, 254, 322	**743**/741	303	1260.0 ± 133.8
11	12.53	Quercetin 3-O-(2, 6-di-*O*-rhamnosyl- glucoside)	256, 352	757/**755**	300, 301, 151	609.4 ± 170.5
12	12.77	Rutin ^a^	254, 352	611/**609**	301	2219.7 ± 342.4
13	13.15	20-Hydroxyecdysone	246, 316, 422	**481**	165, 371, 301, 173	839.7 ± 96.8
14	13.75	Kaempferol 3-*O*-β-rutinoside	228, 266, 316	**595**/593	287	212.5 ± 22.9
15	20.49	Internal standard ^a^	220, 272, 312	**255**/	151, 131, 103, 209	100

^a^ Compound identification by comparison with authentic standards. ^b^ Values in bold indicate the molecular ion for MS/MS fragmentation. ^c^ MS/MS fragment ions are shown with decreasing order according to their signal intensity. *, tentatively identified. Internal standard: 7-methoxyflavanone.

## Data Availability

This study did not report any data.

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
