# Peer review of "Djulis (Chenopodium formosanum) and Its Bioactive Compounds Protect Human Lung Epithelial A549 Cells from Oxidative Injury Induced by Particulate Matter via Nrf2 Signaling Pathway"

_molecules, 2021, doi:10.3390/molecules27010253_

Round 1
Reviewer 1 Report
This manuscript describes a series of studies intended to demonstrate that an aqueous extract of djulis (Chenopodium formosanum) (WECF) and some of its individual bioactive components protect the lung cancer cell line A549 from in vitro oxidative damage and cytotoxicity induced by particular matter (PM). This work is part of a larger body of literature published by the same group describing various positive biological effects of djulis. Although, in principle, this further investigated property (i.e., protection against air pollutants) may arouse interest in the readers, some experimental criticalities considerably weaken the conclusions drawn by the Authors. These are summarized in the following points:
- The Authors use the adenocarcinoma human alveolar basal epithelial cell line A549 to perform the in vitro studies. Have they considered that WECF at doses of 10 μg/ml and 50 μg/ml has a significant effect in promoting the growth of these cancer cells (Fig. 1B)? It does not seem a desirable effect: how do the authors explain this phenomenon? It is believed that for this type of studies it would have been more appropriate to use primary epithelial lung or bronchial cells.
- Figure 1D: this graph is incorrect because the title of the Y axis shows "LDH release (% of control)", but the control histogram has a value of 0 and the PM histogram instead has a presumably value of 100. The correct spelling of the Y axis should be "LDH release (% of PM)".
- The Control bars in Figures 3A, 3B, 3C, 3D, 5G and 5H have standard deviation = 0 (as also reported in the original figures of the western blots in the letter to the Editors), and this is not experimentally possible. The authors should better explain how the calculations of densitometric analyzes of western blots of Nrf2 and HO-1, normalized with respect to beta-actin, were performed.
- Figures 3C and 3D: the western blot images of Nrf2 and HO-1 are different from the corresponding Figg. 3C and 3D shown as original figures in the letter to the Editors.
- Figure 3D: The scheme of this experiment is not clear as the table of treatments with the various compounds below the western blot of beta-actin shows all "-".
- The concentrations of SB203580, SP600125 and U0126 used in the experiments shown in Figures 3C and 3D are not specified in the manuscript. Furthermore, the authors should justify the choice of the single dose used for each of the three inhibitors: were preliminary experiments performed for dose selection? The dose used for each is critical, as they are kinase inhibitors that have a rather promiscuous selectivity profile. More importantly, the effective inhibition of the target kinases of these three compounds (JNK, ERK and p38MAPK) has not been demonstrated following the treatments. It is considered essential to provide kinase inhibition data in cells, for example by analyzing by western blot the phosphorylation of their respective substrates (cJun, p42/p44MAPK, etc).
- Materials and Methods: The catalog numbers of the Cell Signaling Technology anti-Nrf2 and anti-HO-1 antibodies used are missing and must be reported.
- The Discussion section is excessively long: it is recommended to shorten it.
- Among the original images shown in the letter to the Editors, the western blot of beta-actin in figure 3C is identical to the western blot of beta-actin in figure 3D.
Author Response
Dear Reviewer 1
First of all, I would like to thank for your meaningful comments and suggestions. The manuscript has been checked and revised according to the comments or suggestions. The details of the revisions and responses to you are the following:
Comment 1
The Authors use the adenocarcinoma human alveolar basal epithelial cell line A549 to perform the in vitro studies. Have they considered that WECF at doses of 10 μg/ml and 50 μg/ml has a significant effect in promoting the growth of these cancer cells (Fig. 1B)? It does not seem a desirable effect: how do the authors explain this phenomenon? It is believed that for this type of studies it would have been more appropriate to use primary epithelial lung or bronchial cells.
Answer 1
- The cell viability of 10 μg/ml WECF in Fig 1B in original version is slight increase compared to control. The cell viability obtained is 104.2 ±0.6% after the cell viability is conducted again, which no significant differences between the value (104.2 ±0.6%) and the value 111.3±2.9% in original version and the control are found. The different values obtained vary each time, however, no significant difference between them. For 50 μg/ml WECF, the cell viability is 103.07% in original version and no significant difference between 50 μg/ml WECF and control. The label of significant difference in 50 μg/ml WECF has been removed. Please see Fig. 1B.
- The main considerations for the selection of A549 cell in this study are that it has characteristic features of type II cell of pulmonary epithelium, including metabolic and transport capacities. In addition, this cell line is commonly used to investigate a wide range of respiratory ailments. The air born particulates can cause airway inflammation, decline in lung function and incidence and exacerbation of asthma and chronic obstructive pulmonary disease. So A549 cells have been widely used to investigate the effect of PM on lung function. Based on the above reasons, we choose A549 cell in this study.
Comment 2
Figure 1D: this graph is incorrect because the title of the Y axis shows "LDH release (% of control)", but the control histogram has a value of 0 and the PM histogram instead has a presumably value of 100. The correct spelling of the Y axis should be "LDH release (% of PM)".
Answer 2
Many thanks for your comment. LDH release (% of control) has been changed to LDH (% of PM). Please see Fig. 1D.
Comment 3
The Control bars in Figures 3A, 3B, 3C, 3D, 5G and 5H have standard deviation = 0 (as also reported in the original figures of the western blots in the letter to the Editors), and this is not experimentally possible. The authors should better explain how the calculations of densitometric analyzes of western blots of Nrf2 and HO-1, normalized with respect to beta-actin, were performed.
Answer 3
According to the comment 3, the data in Figs 3A, 3B, 3C, 3D, 5G and 5H are recalculated based on the average value of original data. Therefore, the above graphics are remade. Please see Figs 3A, 3B, 3C, 3D, 5G and 5H. Also, the values of control are 1.00 ±0.1, 1.00±0.1, 1.00±0.0, 1.00±0.1, 1.00±0.2, 1.00±0.0 for Figs 3A, 3B, 3C, 3D, 5G and 5H, respectively.
Comment 4
Figures 3C and 3D: the western blot images of Nrf2 and HO-1 are different from the corresponding Fig. 3C and 3D shown as original figures in the letter to the Editors.
Answer 4
The pisels were enhanced in Fig. 3C. However, the data in Fig 3C are the same as the original data those in the original Fig. 3C. For Fig.3D, the band of WECF in the original image was moved from the far right to the third position on the left to facilitate observation and comparison. We confirm that the data in manuscript are correct and not forged.
Comment 5
Figure 3D: The scheme of this experiment is not clear as the table of treatments with the various compounds below the western blot of beta-actin shows all "-".
Answer 5
The description of MAPK inhibitors assay is recruited and added to the Materials and Methods. Please see Lines 654-662.
Comment 6
The concentrations of SB203580, SP600125 and U0126 used in the experiments shown in Figures 3C and 3D are not specified in the manuscript. Furthermore, the authors should justify the choice of the single dose used for each of the three inhibitors: were preliminary experiments performed for dose selection? The dose used for each is critical, as they are kinase inhibitors that have a rather promiscuous selectivity profile. More importantly, the effective inhibition of the target kinases of these three compounds (JNK, ERK and p38MAPK) has not been demonstrated following the treatments. It is considered essential to provide kinase inhibition data in cells, for example by analyzing by western blot the phosphorylation of their respective substrates (cJun, p42/p44MAPK, etc).
Answer 6
1.The concentrations (20 µM in all cases) of three inhibitors have been shown in manuscript, please see Line 657.
- The reason of the single dose used for each of the three inhibitors is based on the references cited and our preliminary experiments tested.
- Indeed, MAPK responds to a variety of extracellular signals and coordinates cellular responses by phosphorylating and regulating the activity of dozens of substrate proteins involved in transcription, translation, and changes in cellular architecture. On the other hand, the ability of manipulate the activity of MAPKs with specific inhibitors has received much attention as research tools for understanding basic mechanism of cellular functions. Although the effects of WECF on MAPK phosphorylation have not been determined, the three MAPKs inhibitors were intervened to elucidate the signaling pathway in this work. We claim that PM-induced oxidative injury in A549 cells is caused by the upregulation of Nrf2 and HO-1 via activation of ERK and JUN signaling pathways. That is to say, the results obtained from intervention of MAPKs inhibitors can at least understand the signaling pathways.
Comment 7
Materials and Methods: The catalog numbers of the Cell Signaling Technology anti-Nrf2 and anti-HO-1 antibodies used are missing and must be reported.
Answer 7
The catalog numbers of anti-Nrf2 (#12721S) and anti-HO-1 (#5853S) have been recruited. Please see Line 643.
Comment 8
The Discussion section is excessively long: it is recommended to shorten it.
Answer 8
The Discussion section has been checked and revised. Some sentences have been modified or deleted. It has been shortened. The following sentences have been deleted:
Lines 305-306, 357-361, 373-376, 397-398 and 442-449 in the original version.
Comment 9
Among the original images shown in the letter to the Editors, the western blot of beta-actin in figure 3C is identical to the western blot of beta-actin in figure 3D.
Answer 9
The image of beta-actin in each test is shown in Fig. 3C and Fig. 3D, respectively. We confirm that each beta-actin in Fig. 3C and Fig. 3D was measured in different times, and it is clearly shown in Fig.3C and Fig.3D, respectively.
Warm regards
Pin-Der Duh
Corresponding author
Reviewer 2 Report
In the present manuscript Chu et.al report the characterization and the bioactivity investigation of Djulis extract to prevent or limit oxidative injury in lung epithelial A549 cells. The manuscript is well-written, and it flows smoothly. The introduction provides a solid background on the Djulis features as well as a comprehensive picture of the PM-induced effects on human cells. The manuscript reports a detailed and solid investigation of the anti-oxidative effects of the Djulis extract along with a composition analysis and biological pathway investigation. The experiments are well-designed and described showing controls and the dose-response results. The conclusions are supported by the results and provide insights on the Djulis extract properties. I do just have few minor points that I would like to be addressed by the authors.
- All the acronyms including, assays, cell lines, chemicals must be explicated the first time they are mentioned along the manuscript facilitate the reader.
- The discussion session is comprehensive, relating the biological causes and pathways to the experimental results in a clear manner. On the other hand, I would suggest the authors to provide a short introduction paragraph or just an introduction sentence on each assay of section 2 to briefly define the investigated effect, and the assay. The results are already clear and very well-analyzed.
- By comparing the data reported in figure 1a and 1c. If I understood correctly at 400 µg/ml of PM the cell viability is 56.6 in 1a and 65.69 in 1c. Is this variation meaningful?
- Rutin and quercitin seems to be the most “valuable” components of the extract showing regulation of the Nrf2 pathway. Can the author comment on the results in comparison to reported one for other extracts?
- As a general analysis, how does the antioxidant features of the Djulis compare to other natural extracts? Do the Djulis show any unique or better property?
Author Response
Dear Reviewer 2
First of all, I would like to thank for your meaningful comments and suggestions. The manuscript has been checked and revised according to the comments or suggestions. The details of the revisions and responses to you are the following:
Comment 1
All the acronyms including, assays, cell lines, chemicals must be explicated the first time they are mentioned along the manuscript facilitate the reader.
Answer 1
The acronyms in assays, cell lines, chemicals in the manuscript have been explicated and indicated at the first time. Please see Lines 18, 19, 21-23, 34-35, 58, 200-201, 253, 579, 583, 594, 610, 638, and 639-640.
Comment 2
The discussion session is comprehensive, relating the biological causes and pathways to the experimental results in a clear manner. On the other hand, I would suggest the authors to provide a short introduction paragraph or just an introduction sentence on each assay of section 2 to briefly define the investigated effect, and the assay. The results are already clear and very well-analyzed.
Answer 2
To provide a short introduction paragraph or just an introduction sentence on each assay of section 2 is a nice suggestion. We have added necessary paragraphs to where they need to be added in the results. Please see Lines 125-127, 146-147, 172-173, 226-228, and 276-277.
Comment 3
By comparing the data reported in figure 1a and 1c. If I understood correctly at 400 µg/ml of PM the cell viability is 56.6 in 1a and 65.69 in 1c. Is this variation meaningful? Rutin and quercitin seems to be the most “valuable” components of the extract showing regulation of the Nrf2 pathway. Can the author comment on the results in comparison to reported one for other extracts?
Answer 3
Because the value 56.6% in Fig. 1A and 65.69% in Fig. 1C is determined in different time, the results vary each time, however, they are all in a meaningful range.
In addition, very few investigations to explore the effects of bioactive compounds on PM-induced lung cells have been found so far. Although the investigation regarding the effects of puerarin, EGCG and statin on PM-induced health risk have been reported. However, no reports have been reported so far on the effectiveness of rutin or quercetin in regulating PM-induced oxidative damage.
Comment 4
As a general analysis, how does the antioxidant features of the Djulis compare to other natural extracts? Do the Djulis show any unique or better property?
Answer 4
Up to now, the biological effects of djulis has not been compared with other natural products or other extracts, but in our previous reports show that djulis in deed showed remarkable biological effects, including hepato-protection. inhibition of hyperglycemia and hyperlipidemia, antiadipogenesis, antihypertension, and anticancer activity.
Warm regards
Pin-Der Duh
Corresponding author
Reviewer 3 Report
Chu et al. addressed that WECF protected against particulate matter (PM)-induced oxidative injury in A549 cells via Nrf2 signaling. WECF is involved in a decrease in ROS generation, TBARS formation, caspase-3 activity, and increases in SOD activity by Nrf2 and HO-1 expression. Although rutin and quercetin in WECF improved PM-induced oxidative stress in the A549 cells, there are some issues that should be addressed before publication.
Major comments
1) The authors tested WECF using only A549 cells, which are lung cancer cells. However, there are numerous normal lung cell lines, such as BEAS-2B, MRC5, and WI-38. It is recommended to use more than one human cell line to test PM-induced oxidative injury and the effects of WECF on normal lung cell lines in Figure 1.
2) The authors showed ROS levels with DCFH-DA indicators. It is recommended that the authors perform flow cytometry analysis with ROS indicators (such as DCFH-DA and MitoSOX) to observe levels of both total and mitochondrial ROS in Figure 2.
3) The authors claim that PM-induced oxidative injury is caused by the upregulation of Nrf2 and HO-1 via the activation of the ERK and JUN signaling pathways. However, the authors used inhibitors of ERK (U0126) and JUN (SP600125) but did not show the protein expression levels of phopho-ERK and JUN via western blotting. Please add the protein levels of phopho- and unphospho-ERK and JUN in Figure 3 to prove that the inhibitors worked properly.
4) The authors showed the effect of WECF on Nrf2 and HO-1 protein expressions in Figure 3. The Nrf2-Keap1 pathway is a protective response to oxidative stress. Kelch-like ECH-associated protein 1 (KEAP1) is a component of the E3 ubiquitin ligase complex and controls the stability and accumulation of NRF2. It is recommended to add western blots of KEAP1 levels under PM treated with the same concentration of WECF in Figure 3A.
5) Apoptosis is triggered by multi-signal pathways and regulated by extrinsic and intrinsic ligands. It would be interesting to explore which apoptotic signal involves particulate matter (PM)-induced oxidative injury. The authors showed the effects of rutin and quercetin on PM-induced caspase-3 activity. It is recommended that the authors perform the experiment with western blots of Caspase-3, cleaved caspase-3, Bax, and Bcl-2 in Figure 5F.
Minor points
1) Please explain the lactate dehydrogenase (LDH) leakage effect related to cytotoxicity in the introduction or discussion section.
2) Please provide a detailed explanation on why WECF treatment can enhance HO-1 levels but rutin and quercetin have little effect on HO-1 in the discussion section.
Author Response
Dear Reviewer 3
First of all, I would like to thank for your meaningful comments and suggestions. The manuscript has been checked and revised according to the comments or suggestions. The details of the revisions and responses to you are the following:
Major comments
Comment 1
1) The authors tested WECF using only A549 cells, which are lung cancer cells. However, there are numerous normal lung cell lines, such as BEAS-2B, MRC5, and WI-38. It is recommended to use more than one human cell line to test PM-induced oxidative injury and the effects of WECF on normal lung cell lines in Figure 1.
Answer 1
Although numerous normal lung cell lines, such as BEAS-2B, MRC5, and WI-38, can be used to test PM-induced oxidative injury, the main reason for selecting A549 cells in this study is that A549 cells, a human lung cell line, have characteristic features of type II cells of pulmonary epithelium, including metabolic and transport capacities. In addition, this cell line is commonly used to investigate a wide range of respiratory ailments. Therefore, we incubated A549 cells with WECF and bioactive compounds followed by PM exposure.
Comment 2
The authors showed ROS levels with DCFH-DA indicators. It is recommended that the authors perform flow cytometry analysis with ROS indicators (such as DCFH-DA and MitoSOX) to observe levels of both total and mitochondrial ROS in Figure 2.
Answer 2
Using flow cytometry analysis to determine both total cell ROS levels and mitochondrial ROS contents is a good suggestion. However, this study focused on the total ROS production in PM-induced A549 cells, thereby evaluating the degree of oxidative damage toward A549 cells. So DCFH-DA was used as an indicator in the present work.
Comment 3
The authors claim that PM-induced oxidative injury is caused by the upregulation of Nrf2 and HO-1 via the activation of the ERK and JUN signaling pathways. However, the authors used inhibitors of ERK (U0126) and JUN (SP600125) but did not show the protein expression levels of phopho-ERK and JUN via western blotting. Please add the protein levels of phopho- and unphospho-ERK and JUN in Figure 3 to prove that the inhibitors worked properly.
Answer 3
Indeed, MAPK responds to a variety of extracellular signals and coordinates cellular responses by phosphorylating and regulating the activity of dozens of substrate proteins involved in transcription, translation, and changes in cellular architecture. On the other hand, the ability of manipulate the activity of MAPKs with specific inhibitors has received much attention as research tools for understanding basic mechanism of cellular functions. Although the effects of WECF on MAPK phosphorylation have not determined, the three MAPKs inhibitors were intervened to elucidate the signaling pathway in this work. We claim that PM-induced oxidative injury in A549 cells is caused by the upregulation of Nrf2 and HO-1 via activation of ERK and JUN signaling pathways. That is to say, the results obtained from intervention of MAPKs inhibitors can at least understand the signaling pathways.
Comment 4
4) The authors showed the effect of WECF on Nrf2 and HO-1 protein expressions in Figure 3. The Nrf2-Keap1 pathway is a protective response to oxidative stress. Kelch-like ECH-associated protein 1 (KEAP1) is a component of the E3 ubiquitin ligase complex and controls the stability and accumulation of NRF2. It is recommended to add western blots of KEAP1 levels under PM treated with the same concentration of WECF in Figure 3A.
Answer 4
Thanks for the reviewer 3 suggestion. KEAP1 is a component of the E3 ubiquitin ligase complex and controls the stability and accumulation of Nrf2. In other words, once Nrf2 is released and translocated to the nucleus, it activates phase II enzymes, such as SOD, γGluCys, HO-1, CAT and GSHPx. So Nrf2 plays a pivotal role in the activation of phase II enzymes. Although KEAP1 protein expression was not measured, the determination of Nrf2 protein expression can at least understand the basic mechanism of signaling pathway. Therefore, we focused the effect of WECF on Nrf2 protein expression in PM-induced A549 cells.
Comment 5
5) Apoptosis is triggered by multi-signal pathways and regulated by extrinsic and intrinsic ligands. It would be interesting to explore which apoptotic signal involves particulate matter (PM)-induced oxidative injury. The authors showed the effects of rutin and quercetin on PM-induced caspase-3 activity. It is recommended that the authors perform the experiment with western blots of Caspase-3, cleaved caspase-3, Bax, and Bcl-2 in Figure 5F.
Answer 5
The aim of this study is to explore the effect of WECF and its bioactive compounds on oxidative damage induced by PM. In previous studies we have found that acceleration or accumulation of ROS generation may promote oxidative damage, thereby leading to apoptosis of cells. Caspase-3, an effector caspase, has been recognized to implement apoptotic steps by cleaving multifarious cellular substrates. Therefore, in addition to exploring the effect of WECF and its bioactive compounds on ROS generation in PM-induced A549 cells, their ability to inhibit apoptosis was also measured, which was evaluated by expression of proteins associated to apoptosis, such as caspase-3 activity. Although the Bax, Bcl-2 and cleaved caspase-3 have not been conducted in this study, we speculate that WECF has a positive regulation and impact on them due to bioactive compounds present in WECF.
Minor points
Comment 1
1) Please explain the lactate dehydrogenase (LDH) leakage effect related to cytotoxicity in the introduction or discussion section.
Answer 1
LDH is widely used as a marker to study the toxicity of toxicants. Fig. 1D shows that A549 cells treated with PM show a significant release of LDH after 24 h, however, a significant decrease in LDH release from cells is observed after exposure to WECF, indicating that WECF prevented PM-induced cell death. This description is added to discussion section. Please see Lines 327-330.
Comment 2
2) Please provide a detailed explanation on why WECF treatment can enhance HO-1 levels but rutin and quercetin have little effect on HO-1 in the discussion section.
Answer 2
The fourteen compounds identified as present in WECF may contribute to enhance Nrf2 and HO-1 protein expression in PM-induced A549 cells due to a direct action or the synergy between the compounds, thereby leading to enhance Nrf2 and HO-1 expression.
In the case of quercetin, it may act as a “double-edged sword” due to its unique properties, since its behave as antioxidants /or pro-oxidant depending on concentration and duration of exposure (Mostafavi‑Pour et al., 2017). That is to say, concentrations and exposure times may be the main factors for quercetin to provoke Nrf2 and HO-1 expression. In the present study, current data show that quercetin, with 1 and 5 µM concentrations and 12 and 15 h exposure times, demonstrated no effect on Nrf2 andHO-1 expression, respectively. Therefore, we speculate that quercetin at 1 and 5 µM co-cultured with PM-induced cells for 12 and 15 h might be not in the appropriate time or concentration to positively regulate the protein expression of Nrf2 and HO-1, respectively. However, this speculation requires further study. This description is added to Lines 460-169.
Reference:
Mostafavi‑Pour, Z., Ramezani, F., Keshavarzi, F., Samadi, N. (2017). The role of quercetin and vitamin C in Nrf2‑dependent oxidative stress production in breast cancer cells. Oncol. Lett. 2017, 1, 1965-1973. https://doi.org/10.3892/ol.2017.5619
Warm regards
Pin-Der Duh
Corresponding author
Round 2
Reviewer 1 Report
The criticisms raised have been sufficiently addressed. It is recommended that this version of the manuscript be considered for publication in Molecules.
Author Response
Dear Reviewer 1
Many thanks for your approving this version of the manuscript for publication in Molecules.
Comments and Suggestions for Authors
The criticisms raised have been sufficiently addressed. It is recommended that this version of the manuscript be considered for publication in Molecules.
Answer
I appreciate for your approval.
Warm regards
Pin-Der Duh
Corresponding author
Reviewer 3 Report
Even though the authors revised according to the comments, there are many issues that should not be addressed properly.
Major comments
1) The authors tested only A549 human lung cell line, type II cells of pulmonary epithelium. To investigate a wide range of respiratory ailments, the effect of WECF and bioactive compounds followed by PM exposure must be tested using more than one lung cell line as a control.
2) The authors claimed that Nrf2 activates phase II enzymes, such as SOD, γGluCys, HO-1, CAT, and GSHPx. So authors must add the target more than two targets such as SOD and CAT after treating with WECF, rutin, and quercetin.
3) The authors speculate that WECF and its bioactive compounds have the ability to inhibit apoptosis. Cleaved caspase-3 propagates an apoptotic signal through enzymatic activity on downstream targets, including poly ADP ribose polymerase (PARP) and other substrates. The authors showed only Caspase-3 activity without additional apoptotic markers. These insufficient data might readers not convincing the effect of PM, WECF, and its bioactive compounds. I strongly suggest that the author must add additional WB data to support inhibiting apoptosis by WECF and its bioactive compounds.
Author Response
Dear Reviewer 3
Many thanks for your comments and suggestions. The manuscript has been checked, revised, and explained according to the comments or suggestions. The details of the revisions and responses to you are the following:
Major comments
Comment 1
The authors tested only A549 human lung cell line, type II cells of pulmonary epithelium. To investigate a wide range of respiratory ailments, the effect of WECF and bioactive compounds followed by PM exposure must be tested using more than one lung cell line as a control.
Answer 1
Indeed, many lung cells may be utilized to study in various model systems, such as
BEAS-2B cell line derived from normal bronchial epithelium, WI-38 or MRC-5 from fetal lung fibroblasts, WTHBF-6 cells, a clonal cell line derived from normal human bronchial fibroblasts, and A549, a malignant cell line and belonged to non-small cell lung carcinoma, which was derived from adenocarcinomic human alveolar basal epithelial cells (A549). These cells are used according to their characteristic features. In addition, lung cancer is the leading cause of cancer-related deaths in the world with non-small cell lung cancer (NSCLC) making up about 85% of all lung cancer cases. Today a large number of human non-small cell lung cancer cell lines exist that are being used for both basic research and drug discovery. Among the cells mentioned above, A549 cells resemble type II cells in a number of important features, because they are readily cultured and derived from a human source, and they are widely used as a model of type II cells. Also, this cell line has been utilized not only for studying lung cancer but also for other infections related to the lungs like allergies, asthma, and respiratory infections. Furthermore, if the cells contain higher levels of metabolic enzymes, which may be due to the fact that they have metabolic capabilities, cannot accurately reflect the oxidative stress of particulates. On the other hand, the human lung epithelial carcinoma cell line (A549) has been widely used to examine the cytotoxic mechanisms resulted from PM2.5 exposures (references cited as below). In addition, epithelial cells play an important role during lung remodeling, which could lead to progression of respiratory diseases. This alveolar epithelial cell line A549 has been used widely and currently accepted experimental model for in vitro inhalation toxicology studies (references cited as below). Furthermore, A549 cells maintain alveolar type II cells characteristics, such as secretion of cytokines, surfactant production and phase I and phase II enzymes for xenobiotic biotransformation similar to lung tissue (Marchetti et al., 2021). Given that the majority of lung cancers are belonged to non-small cell lung cancer (NSCLC), this study focused on non-small cell lung cancer cell line A549 instead of BEAS-2B, WI-38, MRC-5, and WTHBF-6 cells. For these reasons A549 cell line was chosen for this functional research. In addition, our aim was to explored the protective effects of (WECF) and its bioactive compounds toward PM-induced oxidative injury in A549 cells and elucidating the possible mechanisms, so only the terms, human lung epithelial A549 cells, were indicated in the title in this study.
References:
Marchetti, S., Bengalli, R., Floris, P., Colombo, A., Mantecca, P. Combustion‑derived particles from biomass sources differently promote epithelial‑to‑mesenchymal transition on A549 cells. Arch. Toxicol. 2021, 95, 1379–1390.
https://doi.org/10.1007/s00204-021-02983-8
Zhang, K., Nie, D., Chen, M, Wu, Y., Ge, X., Hu, J., Ge, P. Chemical characterization of two seasonal PM2.5. Environments. 2019, 6, 42; doi:10.3390/environments6040042
Zhang, Y., Darland, D., He, Y., Yang, L., Dong, X., and Chang, Y. Reduction of PM2.5 toxicity on human alveolar epithelial cells A549 by tea polyphenol. J Food Biochem. 2018, 42(3): e12496.
Xu, H.,Jiao, X., Wu, Y., Li, S., Cao, L., Dong. L. Exosomes derived from PM2.5‑treated lung cancer cells promote the growth of lung cancer via the Wnt3a/β‑catenin pathway. Oncol. Rep. 2019, 41(2), 1180-1188. https://doi.org/10.3892/or.2018.6862
Akhtar, U.S., Rastogi, N., Mcwhinney, R.D., Urch, B., Chow, C.W., Evans, G.J., Scott, J.A. The combined effects of physicochemical properties of size-fractionated ambient particulate matter on in vitro toxicity in human A549 lung epithelial cells. Toxicol. Rep. 2014, 1, 145–156.
MohseniBandpi, A., Eslami, A., Shahsavani, A., Khodagholi, F., Alinejad, A. Physicochemical characterization of ambient PM2.5 in Tehran air and its potential cytotoxicity in human lung epithelial cells (A549). Sci. Total Environ. 2017, 593–594, 182–190.
Zou, Y., Jin, C., Su, Y., Li, J., Zhu, B. Water soluble and insoluble components of urban PM2.5 and their cytotoxic effects on epithelial cells (A549) in vitro. Environ. Pollut. 2016, 212, 627–635.
Deng, X., Zhang, F., Rui, W., Long, F., Wang, L., Feng, Z., Chen, D., Ding, W. PM2.5-induced oxidative stress triggers autophagy in human lung epithelial A549 cells. Toxicol. In Vitro 2013, 27, 1762–1770.
Deng X, Rui W, Zhang F, Ding W. PM2.5 induces Nrf2-mediated defense mechanisms against oxidative stress by activating PIK3/AKT signaling pathway in human lung alveolar epithelial A549 cells. Cell Biol. Toxicol. 2013, 29, 143–157.
Deng X, Zhang F, Rui W, Long F, Wang L, Feng Z, Chen D, Ding W. PM2.5-induced oxidative stress triggers autophagy in human lung epithelial A549 cells. Toxicol. In Vitro 2013, 27, 1762–1770.
Calcabrini, A., Meschini, S., Marra, M., Falzano, L., Colone, M., De Berardis, B., Paoletti, L., Arancia, G., Fiorentini, C. Fine environmental particulate engenders alterations in human lung epithelial A549 cells. Environ. Res. 2004, 95, 82–91. [CrossRef]
Adamson, I.Y.; Prieditis, H.; Vincent, R. Pulmonary toxicity of an atmospheric particulate sample is due to the soluble fraction. Toxicol. Appl. Pharmacol. 1999, 157, 43–50. [CrossRef] [PubMed]
Bitko, V., and Barik, S. An endoplasmic reticulum-specific stressactivated caspase (caspase-12) is implicated in the apoptosis of A549 epithelial cells by respiratory syncytial virus. J. Cell. Biochem. 2001, 80, 441–454.
Deng, X., et al., 2013. PM2.5 induces Nrf2-mediated defense mechanisms against oxidative stress by activating PIK3/AKT signaling pathway in human lung alveolar epithelial A549 cells. Cell Biol. Toxicol. 29(3), 143-157.
Gualtieri, M., et al. Gene expression profiling of A549 cells exposed to Milan PM2.5. Toxicol. Lett. 2012, 209(2), 136-145.
Montiel-Davalos, A., Alfaro-Moreno, E., Lopez-Marure, R. PM2.5 and PM10 induce the expression of adhesion molecules and the adhesion of monocytic cells to human umbilical vein endothelial cells. Inhal. Toxicol. 2007, 19 (Suppl 1), 91-98.
.
Comment 2
The authors claimed that Nrf2 activates phase II enzymes, such as SOD, γGluCys, HO-1, CAT, and GSHPx. So authors must add the target more than two targets such as SOD and CAT after treating with WECF, rutin, and quercetin.
Answer 2
It is well known that Nrf2, a basic leucine zipper (bZIP), is a major regulator of endogenous antioxidant pathways, which activates and promotes the expression of antioxidant enzymes by entering the nucleus and binding to antioxidant response element (ARE) and then activates the expression of antioxidant enzymes, including catalase (CAT), superoxide dismutase (SOD), nicotinamide adenine dinucleotide phosphate (NADPH) quinone oxidoreductase 1 (NQO1), glutathione S-transferase (GST), and heme oxygenase 1 (HO-1). These enzymes coordinate with each other and play a role in redox regulation (Wang et al., 2021). The reasons that we chose SOD enzyme rather than CAT, GSHPx and other enzymes in addition to HO-1 enzymes in this study are the following.
Among these antioxidant enzymes, SOD is an antioxidant enzyme present in all aerobic cells and is the first line of defense against superoxide anion radicals (.O2−). The presence of .O2− may lead to cancer, cardiovascular disease and aging. Due to the antioxidant capacity of SOD, it has become an important additive for medicines, beverages, health products and cosmetics. Antioxidant enzymes like SOD, CAT, and GSHPx are crucial in scavenging ROS. The .O2− is a typical ROS that is generated in mitochondria, which can change to H2O2 and O2 via the dismutation catalyzed by SOD. H2O2 can further degrade to H2O and O2 by the action of CAT and GSHPx. In other words, SOD is the front line of defense systems, as compared to other antioxidant enzymes, such as CAT and GSHPx. For these reasons, SOD, as a representative of antioxidant enzymes, and HO-1 regulated by Nrf2 were measured in the study. Therefore, Lines 359-365 are rewritten as following, and Lines 390-393 in the last version are deleted:
“In normal physiological conditions, Nrf2 is fixed in the cytoplasm by cytoskeletal protein in kelch like ECH associated protein 1 (keap1). Once Nrf2 is under oxidative stress, it can exert transcriptional activity after trans membrane transferring into nucleus and bind to the antioxidant response element (ARE) promoter sequence or the DNA binding sequence, which subsequently induce the expression of downstream target genes, suc as SOD and HO-1 enzymes [28].”
Reference
Wang, Y., Jin, R., Chen, J., Cao, J., Xiao, J., Li, X., Sun, C. Tangeretin maintains antioxidant activity by reducing CUL3 mediated NRF2 ubiquitination. Food Chem. 2021, 365, 130470. https://doi.org/10.1016/j.foodchem.2021.130470.
Comment 3
The authors speculate that WECF and its bioactive compounds have the ability to inhibit apoptosis. Cleaved caspase-3 propagates an apoptotic signal through enzymatic activity on downstream targets, including poly ADP ribose polymerase (PARP) and other substrates. The authors showed only Caspase-3 activity without additional apoptotic markers. These insufficient data might readers not convincing the effect of PM, WECF, and its bioactive compounds. I strongly suggest that the author must add additional WB data to support inhibiting apoptosis by WECF and its bioactive compounds.
Answer 3
Although the results show that WECF suppresses caspase-3 activity in PM-induced oxidative stress in A549 cells, additional apoptotic markers, like Bax2, BCL-2 and PARP cleavage, are not determined. Truly, only a marker like caspase-3 tested is not very convincing. For these reasons the descriptions of caspase-3 activity and apoptosis as well as the Figure 2E and Figure 5F in the study are deleted.
The followings in the last version are deleted:
Fig. 2E and Fig. 5F. are deleted.
Warm regards
Pin-Der Duh
Corresponding author